# Delay-period activity in frontal, parietal, and occipital cortex tracks noise and biases in visual working memory

Qing Yu[1,2]*, Matthew F. Panichello[3], Ying Cai[4], Bradley R. Postle[1,5], Timothy J. Buschman[3,6]

1 Department of Psychiatry, University of Wisconsin–Madison, Madison, Wisconsin, United States of America, 2 Center for Excellence in Brain Science and Intelligence Technology, Chinese Academy of Sciences, Shanghai, People's Republic of China, 3 Princeton Neuroscience Institute, Princeton University, Princeton, New Jersey, United States of America, 4 Department of Psychology and Behavioral Sciences, Zhejiang University, Hangzhou, People's Republic of China, 5 Department of Psychology, University of Wisconsin–Madison, Madison, Wisconsin, United States of America, 6 Department of Psychology, Princeton University, Princeton, New Jersey, United States of America

* yuqingpsy@gmail.com

**Data Availability Statement:** All data are available from the OSF database (osf.io/ajq3z).

**Funding:** This work was supported by National Institute of Mental Health R01MH064498 to BRP,

## Abstract

Working memory is imprecise, and these imprecisions can be explained by the combined influences of random diffusive error and systematic drift toward a set of stable states ("attractors"). However, the neural correlates of diffusion and drift remain unknown. Here, we investigated how delay-period activity in frontal and parietal cortex, which is known to correlate with the decline in behavioral memory precision observed with increasing memory load, might relate to diffusion and drift. We analyzed data from an existing experiment in which subjects performed delayed recall for line orientation, at different loads, during functional magnetic resonance imaging (fMRI) scanning. To quantify the influence of drift and diffusion, we modeled subjects' behavior using a discrete attractor model and calculated within-subject correlation between frontal and parietal delay-period activity and whole-trial estimates of drift and diffusion. We found that although increases in frontal and parietal activity were associated with increases in both diffusion and drift, diffusion explained the most variance in frontal and parietal delay-period activity. In comparison, a subsequent whole-brain regression analysis showed that drift, rather than diffusion, explained the most variance in delay-period activity in lateral occipital cortex. These results are consistent with a model of the differential recruitment of general frontoparietal mechanisms in response to diffusive noise and of stimulus-specific biases in occipital cortex.

## Introduction

Working memory—the ability to mentally retain and manipulate information to guide behavior—is crucial for many aspects of high-level cognition [1–3]. One prominent neural hallmark of working memory performance is persistent elevated delay-period activity in frontal and

and The Office of Naval Research N000141410681 and National Institute of Mental Health R01MH115042 to TJB. The funders had no role in study design, data collection and analysis, decision to publish, or preparation of the manuscript.

**Competing interests:** The authors have declared that no competing interests exist.

**Abbreviations:** 1O, 1 orientation; 3O, 3 different orientations; AIC, Akaike Information Criterion; BIC, Bayesian Information Criterion; BOLD, blood oxygen level–dependent; DDM, drift–diffusion model; DOM, diffusion-only model; fMRI, functional magnetic resonance imaging; IPS, intraparietal sulcus; LO, lateral occipital cortex; PFC, prefrontal cortex; ROI, region of interest; TR, repetition time.

parietal cortex. Specifically, blood oxygen level–dependent (BOLD) activity in frontal and parietal cortex increases monotonically with memory load and asymptotes at an individual's memory capacity [4,5]. Activity in these networks is thought to reflect the engagement of control [6,7]. For example, one recent study has demonstrated that persistent activity in parietal cortex tracks the demands of binding stimulus content to its trial-specific context, rather than memory load per se [8]. These signals have been shown to correlate with individual memory capacity [4,5] and with memory precision [8–10]. In contrast, persistently elevated activity during the delay period is often absent in the sensory cortex (e.g., occipital cortex for visual information), despite the reliable representation of stimulus-specific information [8,10–13].

Recent psychophysical work has shown that inaccuracies in working memory are due to both random error and systematic biases. For example, when subjects remember features drawn from a uniform stimulus space, their responses are not uniform. Instead, the responses "cluster" around a small number of specific values [14–16]. Further modeling work has demonstrated this clustering can be explained by attractor dynamics that pull memories to specific locations in mnemonic space (e.g., the memory for pink or purple stimuli can be "attracted" to red). Although such drift induces systematic error into mnemonic representations, it also stabilizes them by limiting random diffusion. Importantly, although the magnitude of drift is highest during memory encoding, its continued influence during the ensuing delay period is necessary to counteract the otherwise accumulating effect of random error [16]. Furthermore, engaging attractor dynamics is thought to be especially beneficial when memory load is higher, because increased noise in memory representations can be counteracted by increasing drift toward a few stable representations.

Because load-related imprecision in working memory performance reflects both random diffusion and drift toward stable attractor states, the extent to which each of these factors accounts for load-sensitive delay-period activity in parietal and frontal cortex remains unclear. In the current study, we analyzed data from an existing experiment in which subjects performed delayed recall for line orientation, at different memory loads, during functional magnetic resonance imaging (fMRI) scanning. We modeled subjects' behavior using a discrete attractor model and regressed the resultant load-sensitive estimates of drift and diffusion, estimated across encoding and memory, against load-dependent delay-period activity in parietal and frontal cortex. We found that increases in frontal and parietal delay-period activity were associated with increases in both diffusion and drift, with diffusion explaining more variance. In lateral occipital cortex, in contrast, drift explained more variance than diffusion in delay-period activity. These results provide a novel interpretation of the functions associated with delay-period activity, suggesting that frontoparietal control networks may be engaged to offset load-related diffusive noise, with load-related drift more prominent in occipital cortex.

## Results

### Behavioral performance

Subjects performed a delayed estimation task on line orientations, both inside and outside the MRI scanner ("fMRI sessions" and "behavioral sessions," respectively). On different trials, subjects either remembered 1 orientation (1O), or 3 different orientations (3O). All behavioral results were based on data from the behavioral session, because this session contained an equal number of trials per condition (details in Methods). For subjects who participated in the fMRI sessions ($n = 16$), we first plotted the distribution of raw responses separately for 1O and 3O trials. Recall error, measured as the angular distance between the target orientation and response orientation, increased with increasing memory load, $t(15) = 8.27$, $p = 5.68 \times 10^{-7}$. Furthermore, similar to what has been previously reported for color [14–16], subjects'

responses to orientation working memory also clustered around a small number of orienta-
tions (Fig 1B), which is consistent with previous observations of a repulsive bias away from
cardinal orientations, and an attractive bias toward oblique orientations [17–19].

To account for these clusters, we fit data using a discrete attractor model [16]. This circular
drift–diffusion model (DDM) fits the dynamic evolution of memories with 2 distinct pro-
cesses: random noise (diffusion) and systematic drift toward one of several stable attractors.
Notably, when the drift parameter is removed, the remaining diffusion-only model (DOM) is
equivalent to a classic mixture model [20]. Note that because memory delay was fixed in the
current study, the DDM estimated drift and diffusion across both the encoding and memory
delay periods. We first demonstrated via simulations that the DDM could be successfully fit to
datasets with discrete target values as in the current study: The bias and variance of the esti-
mates were similar across the uniform and discrete condition (S1 Fig), and mean recovery
error did not significantly differ between the uniform and discrete condition for any of the
parameters (all $ps > 0.4$, bootstrap; S2 Fig).

Consistent with previous work on color working memory [16], the DDM provided a better
fit to behavior than the DOM (the average difference in cross-validated log-likelihood across
folds was 3.67 between DDM and DOM). For the DDM, the diffusion and the drift parameters
both increased with memory load ($t[15] = 4.86$, $p = 0.0002$ and $t[15] = 2.43$, $p = 0.028$, respec-
tively), as did the diffusion parameter from the DOM ($t[15] = 6.52$, $p = 9.67 \times 10^{-6}$; Fig 1C).
When we repeated these analyses on a larger set of data ($n = 30$; 16 fMRI subjects and 14
behavior-only subjects), all results were qualitatively similar to those reported here (the average
difference in cross-validated log-likelihood across folds was 6.56 between DDM and DOM).
Critically, the DDM was also better at capturing the classic cardinal bias in orientation data,
compared with the DOM. Pearson correlation between actual and predicted data was positive
and significant for both 1O ($r = 0.840$, $p = 0.00007$) and 3O ($r = 0.876$, $p = 0.000001$) with the
DDM but not with the DOM (1O: $r = -0.080$, $p = 0.745$; 3O: $r = -0.315$, $p = 0.190$; S3 Fig). The
attractor locations estimated by the DDM also clustered around the 2 oblique orientations
(45˚ and 135˚), consistent with the observation in raw data (S4 Fig).

## BOLD signal change in intraparietal sulcus and prefrontal cortex

We next examined the BOLD time course in the intraparietal sulcus (IPS) and in the prefrontal
cortex (PFC) during the working memory task at the 2 memory loads. We observed the classic
pattern of load-sensitive BOLD activity in both regions of interest (ROIs): signal intensity was
sustained above baseline across the delay period in both load conditions (all $ps < 0.001$), with
greater activity for the higher memory load condition (all $ps < 0.01$, including the "late delay"
time point at which the principal BOLD-behavior analyses were carried out; Fig 2A and 2B).

## Modeling load-dependent BOLD activity with behavior at the ROI level

To relate load-dependent BOLD activity in parietal and frontal cortex to behavior, we fit linear
regression models with the parameters of the DDM and subject as the independent variables,
and BOLD activity as the dependent variable. We first used these regression models to calcu-
late within-subject correlations (ANCOVAs) between behavioral parameters (drift and diffu-
sion) and BOLD activity. The results indicated that BOLD activity in both ROIs correlated
significantly with diffusion (IPS diffusion: $r = 0.83$, $p = 0.00004$; PFC diffusion: $r = 0.79$,
$p = 0.0002$) and drift (IPS drift: $r = 0.59$, $p = 0.012$; PFC drift: $r = 0.61$, $p = 0.009$; Fig 2C and
2D).

Next, to evaluate the contribution of drift and diffusion, we found the regression model
that best explained BOLD activity in the 2 ROIs. Comparison between the 4 models of interest

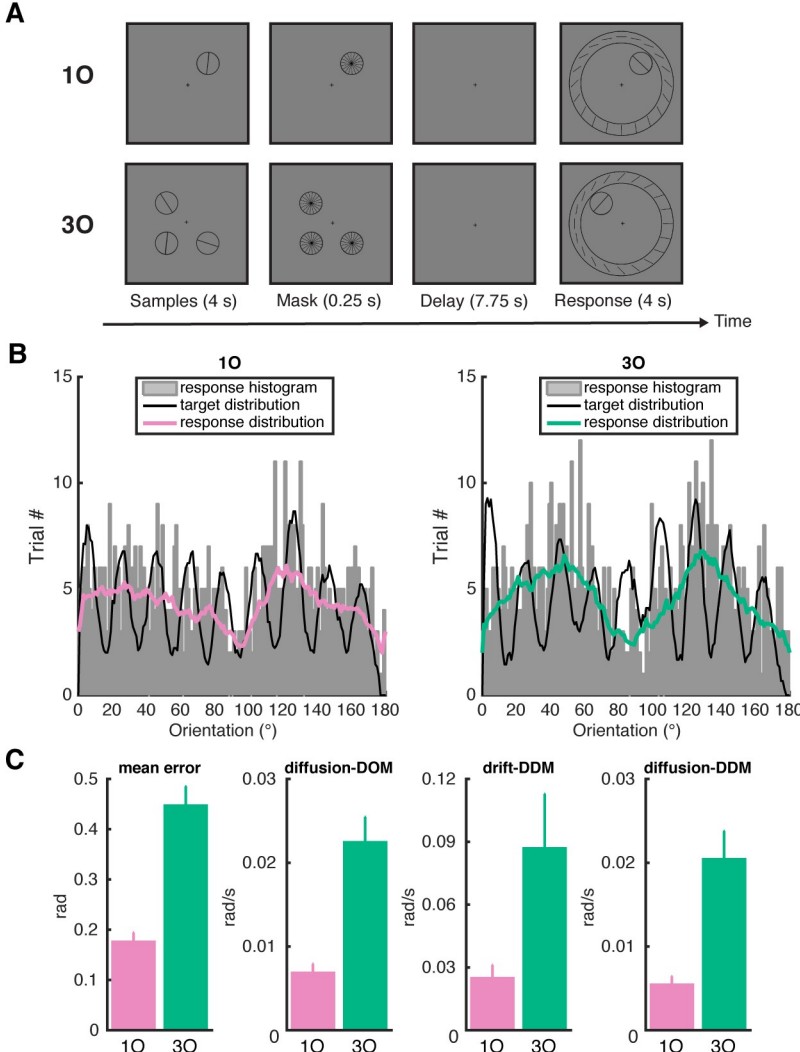

**Fig 1. Trial sequence of the fMRI task and behavioral performance. A**. For the data analyzed in the current study, participants remembered either one or three orientations. Sample stimuli were presented on the screen for 4 seconds, followed by a brief mask period of 0.25 seconds. After a delay of 7.75 seconds, participants rotated the needle of the response wheel to indicate the remembered orientation at the probed location. **B**. The raw response distribution of 1O and 3O trials, indicated by the gray histograms. The black lines indicate the envelope of target distribution, and pink and green lines indicate the envelope of response distribution, for 1O and 3O trials separately. **C**. Model-free and model-based behavioral performance. From left to right panel shows mean error, diffusion from the DOM model, drift from the DDM model, and diffusion from the DDM model. Error bars indicate ± 1 SEM. Data are available at osf.io/ajq3z. 1O, 1 orientation; 3O, 3 different orientations; DDM, drift–diffusion model; DOM, diffusion-only model; fMRI, functional magnetic resonance imaging.

indicated that Model 2 (modeling BOLD activity as a function of diffusion from the DDM) explained the most variance in BOLD activity in both IPS and PFC ROIs, and showed the best model performance in terms of Akaike Information Criterion (AIC), and Bayesian Information Criterion (BIC) (see Table 1 for a complete list of model comparisons).

We also used stepwise regression to examine the relative contribution of drift and diffusion to the prediction of BOLD activity. Starting from Model 3 (modeling BOLD activity as a function of both drift and diffusion from the DDM), stepwise regression removed drift from the

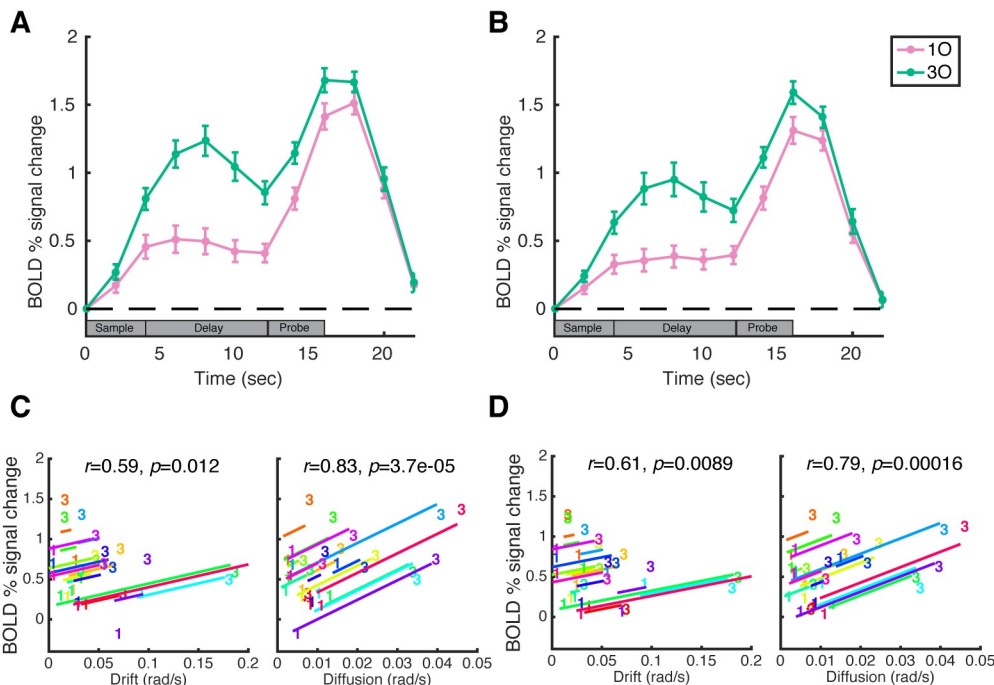

**Fig 2. BOLD activity and brain-behavior correlations in IPS and PFC. A**. Trial-averaged BOLD activity in the IPS functional ROI. **B**. Time course of BOLD activity in the PFC functional ROI. Pink and green lines correspond to the 1O and 3O conditions, respectively. Error bars indicate ± 1 SEM. **C**. Within-subject correlations between behavioral parameter from DDM (drift and diffusion plotted separately) and IPS BOLD activity, at "late delay" time point (12 s). **D**. within-subject correlations between behavioral parameter (drift or diffusion) and PFC BOLD activity. In each plot, data from each subject are plotted in a different color, and the "1" and "3" symbols correspond to values from 1O and 3O trials, respectively. Lines illustrate the best fit of the group-level linear trend (i.e., the within-subject correlation) in relation to individual subject data. Data are available at osf.io/ajq3z. 1O, 1 orientation; 3O, 3 different orientations; BOLD, blood oxygen level–dependent; IPS, intraparietal sulcus; PFC, prefrontal cortex; ROI, region of interest.

model for both IPS ($F[1,14] = 0.35$, $p = 0.564$) and PFC ($F[1,14] = 0.84$, $p = 0.376$) but retained diffusion for both ROIs (diffusion versus constant model: IPS: $F[32,15] = 4.37$, $p = 0.003$; PFC:

**Table 1. Comparison between different regression models.**

| Model | Adjusted $R^2$ | AIC | BIC |
|---|---|---|---|
| IPS | | | |
| Model 1 | 0.237 | 29.379 | 54.297 |
| Model 2 | 0.635 | 5.754 | 30.672 |
| Model 3 | 0.619 | 6.966 | 33.349 |
| Model 4 | 0.580 | 10.255 | 35.176 |
| PFC | | | |
| Model 1 | 0.412 | 14.116 | 39.034 |
| Model 2 | 0.659 | −3.271 | 21.646 |
| Model 3 | 0.652 | −2.817 | 23.566 |
| Model 4 | 0.566 | 4.390 | 29.308 |

AIC, Akaike Information Criterion; BIC, Bayesian Information Criterion; IPS, intraparietal sulcus; PFC, prefrontal cortex; ROI, region of interest.

$F[32,15] = 4.36$, $p = 0.003$). Together, these results suggest the level of BOLD activity in both IPS and PFC is most strongly correlated with the amount of diffusive noise in memories.

## Modeling load-dependent BOLD activity with behavior at the whole-brain level

We next performed a whole-brain linear regression analysis to explore the relative contribution of drift and diffusion to the BOLD activity of each voxel. Consistent with our ROI-based results, we found significant clusters in bilateral IPS and left frontal cortex with load-dependent BOLD activity that can be better explained by load-dependent changes in diffusion (Fig 3A, red clusters). Interestingly, we also observed clusters that showed higher brain-behavior correlation with drift (Fig 3A, green clusters). These clusters were most prominent in the lateral occipital cortex (LO) and superior postcentral gyrus bilaterally and in right inferior precentral gyrus. Because of the known involvement of occipital cortex in visual working memory, we defined 2 bilateral anatomical ROIs for LO (LO1 and LO2) and repeated with them the ROI-based analyses as previously performed for IPS and PFC.

Consistent with previous findings [8,10–13], BOLD signal intensity in the 2 LO ROIs returned to baseline during the delay period, with late-delay-period activity no different from baseline on 1O trials (LO1: $t[15] = 0.300$, $p = 0.868$; LO2: $t[15] = 0.315$, $p = 0.845$) and slightly below baseline on 3O trials (LO1: $t[15] = 2.754$, $p = 0.021$; LO2: $t[15] = 2.369$, $p = 0.043$; Fig 3B and 3C). ANCOVAs between the behavioral parameters from the DDM and this BOLD activity revealed trending correlations with drift (LO1: $r = -0.48$, $p = 0.054$; LO2: $r = -0.44$, $p = 0.081$) and less so with diffusion (LO1: $r = -0.44$, $p = 0.079$; LO2: $r = -0.34$, $p = 0.18$; Fig 3D and 3E). Furthermore, stepwise regression on Model 3 removed diffusion from the model for both LO1 ($F[1,14] = 0.59$, $p = 0.456$) and LO2 ($F[1,14] = 0.13$, $p = 0.727$), whereas drift remained in models for both ROIs (drift versus constant model: LO1: $F[32,15] = 3.98$, $p = 0.005$; LO2: $F[32,15] = 4.2$, $p = 0.004$). This result was opposite of what was observed in the IPS and PFC ROIs.

## Temporal evolution of BOLD-behavior correlations

Because the length of memory delays was not manipulated in this fMRI study, we were limited to obtaining whole-trial behavioral measures of drift and diffusion that did not isolate encoding and delay contributions to these measures. Nonetheless, we were able to investigate whether the brain-behavior correlation changed dynamically over time by examining the temporal evolution of the within-subject correlation between BOLD activity and behavior (S5 Fig and S6 Fig) and also of the model performance of Model 1 and Model 2 (S7 Fig and S8 Fig). We found that in the IPS and PFC, the diffusion model outperformed the drift model, starting from approximately 4 seconds after trial onset, and sustained until approximately 18 seconds after trial onset. Importantly, the difference between the 2 reached its peak around the late-delay period (i.e., approximately 12 seconds after trial onset), the primary focus of our analyses. In comparison, effects in LO1 and LO2 were generally weaker, starting from a diffusion-dominant effect during the sample period (approximately 4–8 seconds after trial onset) and switching to a drift-dominant effect during the delay period. These dynamic changes suggested that our observations with BOLD-behavior correlations for drift and diffusion during delay cannot be interpreted in their entirety as encoding-related effects.

## Discussion

The results of this study provide a new account of the function of load-sensitive activity in the IPS and PFC [4,5]. First, consistent with previous work with color working memory, here we

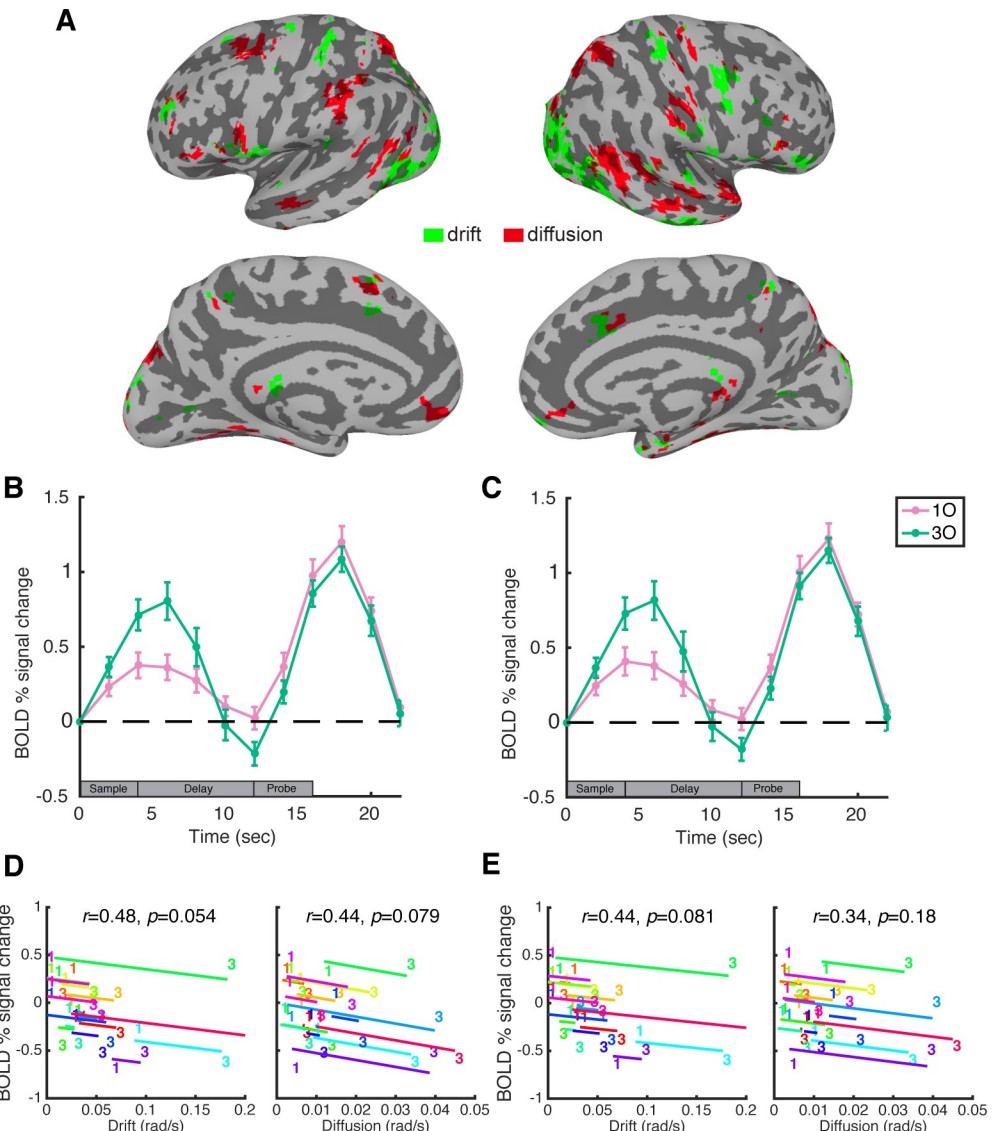

**Fig 3. Whole-brain regression analysis with drift and diffusion and ROI-based results in LO. A**. Whole-brain regression with drift and diffusion. Green denotes voxels showing load-dependent BOLD activity that can be better explained by load-dependent changes in drift, and red denotes voxels showing load-dependent BOLD activity that can be better explained by load-dependent changes in diffusion. For visualization purposes, results were clusterized at a threshold of 20 voxels. The left 2 panels show results from the left hemisphere, and the right 2 panels show results from the right hemisphere. The significance of the regression models was corrected using the FDR method at $p < 0.05$. **B**. Trial-averaged BOLD activity in the LO1 anatomical ROI. **C**. Time course of BOLD activity in the LO2 anatomical ROI. Pink and green lines correspond to the 1O and 3O conditions, respectively. Error bars indicate ± 1 SEM. **D**. Within-subject correlations between behavioral parameter from DDM (drift and diffusion plotted separately) and LO1 BOLD activity, at "late delay" time point (12 seconds). **E**. within-subject correlations between behavioral parameter (drift or diffusion) and LO2 BOLD activity. In each plot, data from each subject are plotted in a different color, and the "1" and "3" symbols correspond to values from 1O and 3O trials, respectively. Lines illustrate the best fit of the group-level linear trend (i.e., the within-subject correlation) in relation to individual subject data. Data are available at osf.io/ajq3z. 1O, 1 orientation; 3O, 3 different orientations; BOLD, blood oxygen level–dependent; FDR, false discovery rate; LO, lateral occipital cortex; ROI, region of interest.

showed that attractor dynamics provided a better account of behavioral data of orientation working memory, compared with classic mixture models that did not take attractor biases into account. Next, and most importantly, the diffusion parameter from the discrete attractor model provided the best account of the load-sensitive delay-period activity of the IPS and PFC. In contrast, in the LO, where aggregate levels of late delay-period activity were at or below baseline levels, load-sensitive fluctuation in this activity was better explained by drift, although the effect was comparably weaker. Thus, our results provide the first evidence to our knowledge that load-related increases in random diffusion during memory delay, one of the important factors that determine imprecision in working memory, engages control-related circuits of the IPS and PFC. Drift toward stable attractor states, on the other hand, may be more important in sensory-related circuits of the LO.

By definition, working memory is guided by information specific to the current trial. Nevertheless, working memory is also often influenced by many other factors, such as sensory history [21] and prior knowledge. In working memory for color, the influence of prior knowledge is reflected as clustered responses around a small number of specific color values, even when the distribution of sample colors is uniform [14–16]. The present results show that this phenomenon generalizes to another low-level visual feature (orientation), and these biases increased with increasing memory load. Together with those of the previous study [16], our results indicate that dynamical systems modeling offers a useful framework within which to understand the influence of trial-nonspecific factors on working memory performance. This framework is also compatible with other models of working memory for orientation, such as the neural resource model [17], in characterizing the well-known response biases for cardinal and oblique orientations [18,19].

Neurally, delay-period neural activity in the IPS and PFC increased with increasing memory load, and we observed that this load-dependent change in BOLD activity was more related to load-dependent changes in diffusion than in drift. Therefore, load-related activity change in the IPS and PFC is more likely related to random diffusion processes. The random noise could be related to noise in representations when memories are held in the IPS/PFC or related to greater engagement of control processes when working memory has greater diffusion. For example, a recent study has found that delay-period activity in IPS is more sensitive to the demands of context binding than of memory load, per se [8]. By this account, increases in diffusion could be due, at least in part, to increased interference between representations of stimulus content and stimulus context, which would be expected to place greater demands on a frontoparietal priority map controlling visually guided behavior.

In comparison to the IPS and PFC, although the effects were generally weaker, delay-period activity in the LO showed an opposite trend, in that it was more sensitive to load-related changes in drift to particular stimulus values (i.e., to attractor strength) than it was to load-related changes in diffusion. This result is consistent with other demonstrations of bias in visual cognition, such as the influence of prior expectation on the representation of motion [22] and the influence of learning on representations of category boundaries in visual cortex [23]. Because our current study did not involve a learning intervention and did not focus on stimulus-specific representations, our results suggest that what we already know about the neural bases of the biasing effects of recent experience on visual cognition may extend to more trait-like "preexisting" attractor landscapes that have been sculpted through a life-time of experience with the visual world.

When considering these findings, it is important to not think of these factors as working in isolation. In frontoparietal cortex, for example, inclusion of drift in the behavioral model provides a better prediction of neural signals with diffusion in these regions. Furthermore, it is important to note that the correlation between load-sensitive drift and BOLD activity in

frontoparietal cortex was also significant; diffusion was simply the parameter that explained more variance in BOLD data. Therefore, it is possible that occipital and frontoparietal regions are both implicated in the generation and processing of drift and diffusion in human working memory performance but that the functioning of frontoparietal cortex is less influential on load-related increases in drift than it is on load-related increases in diffusion. Bias-related processes in frontoparietal and occipital cortex may also play different roles in behavior, with the former reflecting control processes related to implementation of biases, and the latter reflecting stimulus-specific representations that are biased toward attractors. Lastly, drift-dominant effects in LO were comparably weaker and were only present during part of the trial (the delay and part of the probe period), compared with the larger-in-magnitude, trial-spanning diffusion-dominant effects in frontoparietal cortex. Although this difference cannot by simply explained by a difference in BOLD activity between loads, as seen in the comparison of time courses of correlations, it remains to be examined whether this pattern would change, if, for example, a higher memory load was employed to increase load-dependent effect sizes in BOLD and behavior.

Finally, because our study did not manipulate the length of the encoding or delay periods, unique influences from encoding- and delay-related processes on our estimates of drift and diffusion parameters cannot be dissociated. Therefore, our results should not be interpreted as being specific to the delay period but rather as illustrating relations between delay-period neural activity and whole-trial measures of drift and diffusion. Because previous behavioral work has suggested a substantially weaker influence of the delay period on memory performance, compared with encoding [24], one might ask whether the whole-trial measures reported here might also be dominated by encoding-related processes. Although overall magnitude of drift has, indeed, been shown to be higher during encoding [16], we nonetheless believe that delay-related processes had an important contribution to our estimates of drift and diffusion, for at least 2 reasons. First, because drift and diffusion during memory exert their effects continuously over time, their influence on behavior can substantially accumulate over the course of delay periods. For example, in the previous study [16], delay-period drift has been shown to account for the increase in mean absolute bias on long versus short delay-period trials. Second, from a functional perspective, drift counteracts the tendency of random diffusion to accumulate over time [16,25], and this countervailing influence of drift should scale with the rate of diffusion. Indeed, the previous study shows the lower drift rate during memory is sufficient to substantially prevent the accumulation of random errors over time [16]. An important goal for future research will be to systematically manipulate demands on encoding versus memory delay, to directly measure the neural activity attributable to drift and diffusion during encoding versus during the ensuing delay.

In previous studies emphasizing stimulus-specific representations of visual working memory, we have argued that disparate patterns of results in frontoparietal versus occipital cortex are consistent with a functional distinction between these 2 regions, with the former more strongly associated with control and the latter with stimulus representation [8,10]. Here, we see that stimulus-nonspecific factors, as reflected in the relationship between load-dependent changes in behavior (drift and diffusion) and delay-period activity, are also suggestive of such a distinction. Taken together, our results suggest imprecision in working memory is due to a combination of stimulus-related biases in occipital cortex and random diffusion that engages higher-order frontal and parietal cortex.

## Methods

### Ethics statement

This study was approved by the University of Wisconsin–Madison Health Sciences Institutional Review Board (2017–0344) and was conducted according to the principles of the Declaration of Helsinki. Participants provided written informed consent prior to participation.

## Subjects

The results reported here are from analyses carried out on existing data collected for other purposes [26,27]. Thirty individuals (mean age 20.7 ± 2.3 years, 10 males) participated in the behavioral session of the study, and 16 of these (mean age 20.6 ± 1.8 years, 8 males) also participated in 2 subsequent fMRI scanning sessions. All were recruited from the University of Wisconsin–Madison community. All had normal or corrected-to-normal vision and reported no neurological or psychiatric disease. Anatomical scans from the fMRI session were also screened by a neuroradiologist, and no abnormalities were detected. All subjects were monetarily compensated for their participation.

## Stimuli and procedure

All stimuli were created and presented using MATLAB (MathWorks, Natick, MA; www. mathworks.com) and Psychtoolbox 3 extensions (psychtoolbox.org) [28,29]. In the behavioral session, stimuli were presented at a viewing distance of 62 cm on an iMac screen, with a refresh rate of 60 Hz. Subjects registered behavioral responses on a trackball response pad. In the fMRI session, stimuli were projected onto a 60-Hz Avotec Silent Vision 6011 projector (Avotec, Stuart, FL), and viewed through a coil-mounted mirror in the MRI scanner at a viewing distance of 69 cm. Subjects registered behavioral responses on a MR-compatible trackball response pad (Current Designs Inc., Philadelphia, PA).

There were 3 types of stimuli: oriented bars, color patches, or luminance patches. Each oriented-bar stimulus appeared as a black line (width = 0.08˚) bisecting a white circle (radius = 2˚). Line orientations were drawn from a pool of 9 orientations ranging from 0˚ to 160˚, in 20˚ increments, with a random jitter of ±0˚–5˚ added to stimulus on each trial, and another random, fixed jitter of 1˚–10˚ to each participants' 9 target orientations. Color patches were circular patches (radius = 2˚) filled with 1 color drawn from a pool of 9 colors that were equidistant in CIEL*a*b color space (L = 70, a = 20, b = 38, radius = 60˚), with a random jitter of 1˚–5˚. Luminance patches were rendered as a gray circular patch (radius = 0.83˚) inside a white annulus (radius = 2˚), and the luminance of the patches were drawn from 9 grayscale values from (0.03, 0.03, 0.03) to (0.97, 0.97, 0.97), in steps of 0.1175. Throughout the experiment, the background screen color was gray (0.5, 0.5, 0.5).

There were 3 different trial types. On "1O" trials, 1 oriented bar was presented at 1 of 4 possible locations (45˚, 135˚, 225˚, 315˚ relative to central fixation, with an eccentricity of 5˚) for 4 seconds. Stimulus offset was followed by a mask (white circle [radius = 2˚] bisected by 18 black bars [width = 0.08˚] intersecting at their midpoints and each differing in orientation from its neighbors by 10˚; 0.25 seconds) and a delay period (7.75 seconds) during which subjects maintained central fixation. Recall was prompted by the onset of a stimulus circle appearing at the same location as the sample, a response wheel centered on fixation (inner radius = 7.2˚, outer radius of 9.2˚), and a cursor (a conventional "mouse" arrow) located at central fixation. Twenty oriented lines (radius = 1.8˚, width = 0.05˚, ranging in orientation from 0˚ to 171˚ in steps of 9˚) were displayed with equal spacing along the response wheel, and subjects registered their memory of the sample orientation by moving the cursor to the appropriate location on the response wheel and registering that location with a button press. At the onset of the recall display, the stimulus patch was rendered with a randomly determined value rendered in the format of the sample stimuli, and as soon as the subject began to move the cursor (with the trackball) the stimulus patch took on the value corresponding to the location on the response wheel that was nearest to the cursor. Responses were required within 4 seconds, while the circle and wheel remained on the screen. The angle of rotation of the response wheel was

randomized across trials, to prevent subjects from preparing their response during the delay period.

"3O" trials were similar to "1O" trials, except 3 oriented bars, each with a different orientation, were displayed in 3 of the 4 possible sample locations, and at time 12 seconds, the sample to be recalled was indicated by the location of the stimulus circle in the recall array. For each 3O trial, sample values were selected randomly, without replacement, from the pool of 9 possible orientations (Fig 1A).

On "1O1C1L" trials, 1 oriented bar, 1 color patch, and 1 luminance patch were presented, and during the response stage, subjects were tested, unpredictably, on their memory for 1 of these stimuli. The response wheel for color and luminance was the same size as the orientation wheel but displayed 180 possible color or luminance values.

The behavioral session contained 2 blocks of 1O and 3O trials and 3 blocks of 1O1C1L trials. Each block contained 50 trials, and block order was counterbalanced across subjects. The 1O and 3O blocks contained 25 trials each for 1O and 3O, and the 1O1C1L blocks contained 17 probes of 2 of the 3 categories, and 16 of the remaining one. The selection of the categories was randomized across blocks, yielding 50 trials for each category across 3 blocks.

There were 2 fMRI scanning sessions. The first scanning session included four 18-trial blocks of 9 3O trials and 9 1O1C1L trials (with 3 probes each for orientation, color, and luminance), yielding a total of 36 trials for each of these load-of-3 trial types. These 4 blocks were followed by eight 18-trial blocks of 1O trials. The second session included 12 blocks of 1O trials. To match the number of trials between conditions in fMRI data, 2 of the twenty 1O blocks were randomly selected for each subject for further analyses.

We introduce the 1O1C1L condition here only for the completeness of experimental design. All subsequent analyses focused on 1O and 3O trials for load-related changes in behavioral and neural data.

## Behavioral modeling

We fit the data from the behavioral session using a discrete attractor model [16]. This model assumed that memories evolve over time according to 2 distinct processes: random noise (diffusion) and systematic drift toward attractor states in the stimulus space. Specifically, the temporal evolution of a remembered stimulus orientation $\theta$ is modeled using the partial differential equation:

$$d\theta = \beta_L G(\theta)dt + \sigma_L dW$$

where $G$ is a function describing the direction and magnitude of drift across stimulus space, $\beta_L$ defines the gain of the drift for a given load $L$, and $W$ is an additive white noise process with variance $\sigma_L$. Thus, $\beta_L G(\theta)dt$ describes the influence of drift and $\sigma_L dW$, the influence of random noise on memory. By analogy with the decision-making literature, in which drift–diffusion models are often used to model latent evidence for a behavioral choice in a linear 1-D space, this DDM describes the latent value of the subject's memory in a circular 1-D space. Both the drift ($\beta_L$) and diffusion ($\sigma_L$) parameters are rates, with a unit of rad/s indicating the rate of diffusion and the maximum instantaneous drift rate. Unlike the previous study [16], here we fit behavioral data without separating out encoding and delay processes, because the length of memory delays was not manipulated in this experiment.

The model also captures variance due to random responses and reports of nontarget items (following [20,30], see [16] for details). We identified the maximum likelihood estimates of $\beta_L$ (drift) and $\sigma_L$ (diffusion) for each subject and analyzed these fit parameters as described in the main text. Notably, when the drift parameter is removed, the remaining DOM is equivalent to

a classic mixture model [20]. The comparison between performance of the DDM and DOM models was evaluated by combining data from all subjects and computing the average difference in cross-validated log-likelihood value across 10 folds.

## Model simulation on discrete target values

To illustrate whether the DDM could be successfully fit to datasets with discrete target values, we generated simulated data from the DDM using known parameters and tested the ability of our fitting procedure to recover these parameters when targets in these simulated datasets were discrete and uniform. Simulated datasets were matched to the experimental behavioral datasets: each consisted of 50 1O and 50 3O trials with an 8-second delay. The orientation of targets and nontargets were either (1) drawn from a pool of 9 orientations ranging from 0˚ to 160˚ in 20˚ increments with a random jitter of 1˚–10˚ added to each participant's targets and ± 0˚–5˚ added to stimulus on each trial (discrete condition) or (2) drawn uniformly from values ranging from 0˚ to 180˚ (continuous condition). The parameters of the model were set to be similar to those observed empirically: 1O drift, 1O diffusion, 3O drift, and 3O diffusion were set to 0.04, 0.01, 0.10, and 0.03, respectively. The drift function was parameterized such that values were biased toward approximately 45˚ and 135˚, consistent with our observations here and with previous reports in the literature. For each trial, a simulated report was drawn from the probability distribution over reports generated by the model (given the chosen parameters, target value, nontarget values, delay, and set size).

A total of 1,000 discrete and 1,000 continuous datasets with randomly generated target and nontarget values were created. For each dataset, we fit the DDM model and recovered the maximum likelihood parameters, exactly as for the empirical data, and compared the recovered parameters to the known, generative parameters (S1 Fig and S2 Fig).

A similar simulation procedure was also used, based on parameters from model fits of each subject, when plotting the model fits in S3 Fig, to simulate a larger number of trials for more stable model performance (trial number = 1,000).

## fMRI data acquisition

Whole-brain images were acquired with a 3 Tesla GE MR scanner (Discovery MR750; GE Healthcare, Chicago, IL) at the Lane Neuroimaging Laboratory at the University of Wisconsin–Madison HealthEmotions Research Institute (Department of Psychiatry). Functional images were acquired with a gradient-echo echo-planar sequence (2-second TR, 25-millisecond echo time [TE], 60˚ flip angle) within a 64 × 64 matrix (40 sagittal slices, 3.5-mm isotropic). Each of the fMRI scans generated 215 volumes. A high-resolution T1 image was also acquired for each session with a fast-spoiled gradient-recalled-echo sequence (8.2-millisecond TR, 3.2-millisecond TE, 12˚ flip angle, 172 axial slices, 256 × 256 in-plane, 1.0 mm isotropic).

## fMRI data preprocessing

Functional MRI data were preprocessed using AFNI (afni.nimh.nih.gov) [31]. The data were first registered to the first volume of the first run and then to the T1 volume of the first scan session. Six nuisance regressors were included in GLMs to account for head motion artifacts in 6 different directions. The data were then motion corrected, detrended (linear, quadratic, cubic), converted to percent signal change, and spatially smoothed with a 4-mm FWHM Gaussian kernel. For the whole-brain analysis, the data were further aligned to the MNI-ICBM 152 space [32].

## ROI definition

We first defined anatomical ROIs using existing anatomical atlases and warped them back to each subject's structural scan in native space. Parietal anatomical ROIs were created by extracting IPS masks IPS0-5 from the probabilistic atlas of Wang and colleagues [33], merging them, and collapsing over the right and left hemispheres. Lateral PFC anatomical ROIs were created by extracting masks of the superior, middle, and inferior frontal gyri supplied by AFNI, merging them, and collapsing over the right and left hemispheres. Lateral occipital anatomical ROIs were created by extracting masks for LO1 and LO2 from the probabilistic atlas of Wang and colleagues [33], merging them, and collapsing over the right and left hemispheres.

To find the functionally activated voxels within the anatomical atlases, a conventional mass-univariate general linear model (GLM) analysis was implemented in AFNI, with sample, delay, and probe periods of the task modeled with boxcars (4 seconds, 8 seconds, and 4 seconds in length, respectively) that were convolved with a canonical hemodynamic response function. Across the whole brain, we identified the 2,000 voxels displaying the strongest loading on the contrast (delay–baseline), collapsing over all 3 conditions. The intersection of these 2,000 voxels and the 2 anatomical masks defined the 2 functional ROIs in subsequent analyses: the IPS ROI and the PFC ROI. On average, the IPS functional ROI contained 463 ± 177 voxels, the PFC functional ROI contained 314 ± 86 voxels, and the 2 anatomical LO ROIs contained 404 ± 57 and 456 ± 69 voxels, respectively.

## Univariate analyses

We calculated the percent signal change in BOLD activity relative to baseline for each time point during the working memory task; baseline was chosen as the average BOLD activity of the first TR of each trial. The BOLD signal change was averaged across trials within each condition, and across all voxels within each ROI. Statistical significance of BOLD activity against baseline was assessed using 2-tailed, 1-sample t-tests against 0, and the obtained $p$-values were corrected across loads and time points using false discovery rate (FDR) [34]. Statistical difference of BOLD activity between 1O and 3O at each time point was assessed using 2-tailed paired t-tests, and similarly, the obtained $p$-values were FDR corrected across time points.

## Brain-behavior correlation and model comparisons

Following previous work [8–10], we used an analysis of covariance (ANCOVA) method to evaluate the correlated sensitivity to trial type (i.e., 1O versus 3O) across pairs of task-related variables (i.e., BOLD activity versus behavioral parameter). Unlike simple correlations, ANCOVA accommodates the fact that each subject contributes a value for each level of trial type. It removes between-subject differences and assesses evidence for "within-subject correlation" between the 2 task-related variables [35].

Mathematically, within-subject correlations were implemented as linear regression models and were calculated for *drift* and *diffusion* separately, where *subject* is a dummy variable for trial types (1O and 3O) of each subject, and *BOLD* is BOLD signal from time 12 second ("late delay-period" activity), $\beta$s are the regression coefficients, and $\varepsilon$ is the error term.

$$\text{Model 1}: \ BOLD = \beta_2 \times drift_{DDM} + \beta_1 \times subject + \beta_0 + \varepsilon;$$

$$\text{Model 2}: \ BOLD = \beta_2 \times diffusion_{DDM} + \beta_1 \times subject + \beta_0 + \varepsilon;$$

The within-subject correlation *r* for drift or diffusion was calculated as follows:

$$r = \frac{\sqrt{SS_{\text{drift or diffusion}}}}{\sqrt{SS_{\text{drift or diffusion}} + SS_{\text{error}}}}$$

where *SS* stands for sum of squares.

To compare between the performance of different regression models, we included 2 more models, one full model that took both drift and diffusion into account, and one control model that used diffusion from the DOM model:

$$\text{Model 3}: \ BOLD = \beta_3 \times drift_{DDM} + \beta_2 \times diffusion_{DDM} + \beta_1 \times subject + \beta_0 + \varepsilon;$$

$$\text{Model 4}: \ BOLD = \beta_2 \times diffusion_{DOM} + \beta_1 \times subject + \beta_0 + \varepsilon.$$

Model performance was evaluated by comparing AIC, BIC, and adjusted $R^2$ (explained variance of the model after adjusting for the number of predictors) of each model.

Lastly, we performed stepwise regression to evaluate the contribution of the drift and diffusion parameters to the prediction of BOLD activity. The regression model started with Model 3, after the initial fit, the predictors in the model were examined one by one, and the predictor with a $p > 0.10$ in the *F*-test after removal was removed.

### Whole-brain regression analysis

To explore brain areas that showed activity sensitive to either the drift or diffusion parameter, we used a whole-brain exploratory analysis to find voxels with activity that can be best explained by either drift or diffusion. To this end, all subjects' data were first normalized to the MNI-ICBM 152 space [32], and for each voxel, we fit Models 1 and 2 to the BOLD activity of that voxel. The model with a higher adjusted $R^2$ for each voxel was selected as the best fitting for that voxel, and we used the *p*-value of the selected model (*F*-test on regression versus constant model) for statistical significance. To correct for multiple comparisons, we applied the FDR method to the *p*-values of the selected model across voxels. To avoid overinterpretation, we also applied a threshold in model selection using BIC [36], such that only voxels with a significant *p*-value after correction, and in which the drift or diffusion model outperformed the other by a BIC $\geq 2$, remained in the final report. Therefore, we identified voxels with load-dependent BOLD activity that could be better explained by load-dependent changes in drift, or in diffusion, at the whole-brain level. Results from the whole-brain analysis were displayed on the cortical surface reconstructed with FreeSurfer (surfer.nmr.mgh.harvard.edu; [37,38]) and visualized with SUMA in AFNI (afni.nimh.nih.gov) [31].

### Supporting information

**S1 Fig. Distribution of recovered parameters across 1,000 datasets (histogram) and generative values (lines).** Pink: 1O condition. Green: 3O condition. Note that although we observe some bias in the estimate of drift when load = 3 due to the relatively modest number of trials per condition (50), the models fit to the empirical data nevertheless capture behavioral well (S3A Fig) and because this bias is a constant factor in our analyses across regions it cannot explain our neural results. Data are available at osf.io/ajq3z. 1O, 1 orientation; 3O, 3 different orientations.
(TIF)

**S2 Fig. Mean recovery error for known generative parameters of the DDM (values in parentheses).** Violin plots show distribution over 1,000 simulated datasets. Red crosses

indicate mean values. Pink: 1O condition. Green: 3O condition. Data are available at osf.io/ajq3z. 1O, 1 orientation; 3O, 3 different orientations; DDM, drift–diffusion model.
(TIF)

**S3 Fig. Response bias as a function of sample orientation for 1O and 3O conditions (*n* = 30).** Sample orientations were categorized in 10˚ bins. Solid lines demonstrate the experimental data (shaded areas indicate ± 1 SEM), and dashed lines demonstrate model fits. **A.** behavioral data with DDM model fits. **B.** behavioral data with DOM model fits. Data are available at osf.io/ajq3z. 1O, 1 orientation; 3O, 3 different orientations; DDM, drift–diffusion model; DOM, diffusion-only model.
(TIF)

**S4 Fig. Distribution of fitted attractor locations (in 20˚ bins; *n* = 30).** Data are available at osf.io/ajq3z.
(TIF)

**S5 Fig. Time course of difference in BOLD activity between 1O and 3O (3O – 1O, gray) and of within-subject correlation for drift (orange) and diffusion (blue; correlations are shown in absolute values for comparisons).** Error bars indicate ± 1 SEM. **A.** IPS. **B.** PFC. **C.** LO1. **D.** LO2. Data are available at osf.io/ajq3z. 1O, 1 orientation; 3O, 3 different orientations; IPS, intraparietal sulcus; LO, lateral occipital cortex; PFC, prefrontal cortex.
(TIF)

**S6 Fig. Same as S5 Fig, except that difference in correlation (diffusion–drift, purple) was shown.** Positive difference in correlation indicates higher correlation for diffusion, and negative difference indicates higher correlation for drift. **A.** IPS. **B.** PFC. **C.** LO1. **D.** LO2. Data are available at osf.io/ajq3z. 1O, 1 orientation; 3O, 3 different orientations; IPS, intraparietal sulcus; LO, lateral occipital cortex; PFC, prefrontal cortex.
(TIF)

**S7 Fig. Time course of difference in BOLD activity between 1O and 3O (3O – 1O, gray) and of explained variance for Model 1 (orange) and Model 2 (blue).** Error bars indicate ± 1 SEM. **A.** IPS. **B.** PFC. **C.** LO1. **D.** LO2. Data are available at osf.io/ajq3z. 1O, 1 orientation; 3O, 3 different orientations; IPS, intraparietal sulcus; LO, lateral occipital cortex; PFC, prefrontal cortex.
(TIF)

**S8 Fig. Same as S7 Fig, except that difference in explained variance (Model 2 –Model 1, purple) was shown.** Positive difference in explained variance higher model fit for Model 2 (diffusion model), and negative difference indicates higher model fit for Model 1 (drift model). **A.** IPS. **B.** PFC. **C.** LO1. **D.** LO2. Data are available at osf.io/ajq3z. 1O, 1 orientation; 3O, 3 different orientations; IPS, intraparietal sulcus; LO, lateral occipital cortex; PFC, prefrontal cortex.
(TIF)

## Author Contributions

**Conceptualization:** Qing Yu, Matthew F. Panichello, Bradley R. Postle, Timothy J. Buschman.

**Data curation:** Qing Yu, Ying Cai.

**Formal analysis:** Qing Yu, Matthew F. Panichello.

**Funding acquisition:** Bradley R. Postle, Timothy J. Buschman.

**Investigation:** Ying Cai.

**Methodology:** Qing Yu, Matthew F. Panichello.

**Resources:** Qing Yu, Matthew F. Panichello, Ying Cai, Bradley R. Postle.

**Software:** Qing Yu, Matthew F. Panichello, Ying Cai.

**Supervision:** Bradley R. Postle, Timothy J. Buschman.

**Validation:** Qing Yu, Matthew F. Panichello.

**Visualization:** Qing Yu, Matthew F. Panichello.

**Writing – original draft:** Qing Yu.

**Writing – review & editing:** Qing Yu, Matthew F. Panichello, Bradley R. Postle, Timothy J. Buschman.

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
