## [Editor Report · Decision Letter 0]

13 Feb 2020

Dear Dr Yu, 

Thank you for submitting your manuscript entitled "Delay-period activity in frontal, parietal, and occipital cortex tracks different attractor dynamics in visual working memory" for consideration as a Short Report by PLOS Biology.

Your manuscript has now been evaluated by the PLOS Biology editorial staff, as well as by an Academic Editor with relevant expertise, and I am writing to let you know that we would like to send your submission out for external peer review.

Please re-submit your manuscript within two working days, i.e. by Feb 18 2020 11:59PM.

Kind regards,

Gabriel Gasque, Ph.D.,

Senior Editor

PLOS Biology

---

## [Decision Letter · Decision Letter 1]

31 Mar 2020

Dear Dr Yu,

Thank you very much for submitting your manuscript "Delay-period activity in frontal, parietal, and occipital cortex tracks different attractor dynamics in visual working memory" for consideration as a Short Report at PLOS Biology. Your manuscript has been evaluated by the PLOS Biology editors, by an Academic Editor with relevant expertise, and by three independent reviewers. You will note that reviewers 1 and 2, Edward Ester and Edward Vogel, have signed their comments.

The reviews of your manuscript are appended below. As you will see, the referees raise significant technical concerns that could impact the interpretation of results. In addition, they see a need for additional controls or control analyses to rule out potential confounds of design (such as sampling from nine orientations with minor jitter), as studies have shown that participants can acquire long-term memories representations of memoranda. Also, reviewer 3 thinks the study has major limitations as the drift diffusion model is applied to fit data from a single time point and refers to other studies that show drift or bias that don't involve working memory. While we are open to consider a revision that addresses these points, we would only be able to pursue eventual publication if these are addressed exhaustively and you manage to persuade our reviewers about the robustness of the study. We cannot make any decision about publication until we have seen the revised manuscript and your response to the reviewers' comments. Your revised manuscript would be sent for further evaluation by the reviewers.

We appreciate that these requests represent a great deal of extra work, and we are willing to relax our standard revision time to allow you six months to revise your manuscript.We expect to receive your revised manuscript within 6 months.

**IMPORTANT - SUBMITTING YOUR REVISION**

Your revisions should address the specific points made by each reviewer. Because this is a Short Report, please keep the number of main figures to a maximum of four.

Please submit the following files along with your revised manuscript:

*Resubmission Checklist*

*Published Peer Review*

*PLOS Data Policy*

*Blot and Gel Data Policy*

Sincerely,

Gabriel Gasque, Ph.D., 

Senior Editor

PLOS Biology

REVIEWS:

Reviewer #1: Synopsis: Yu et al. use a novel modeling approach to estimate parameters for bias and variance in an orientation recall task, then examine how these parameters co-vary with memory-load-dependent changes in frontoparietal BOLD activity during the delay period of the task. They find that load-dependent changes in frontoparietal activity are correlated with both variance and bias, with the former explaining a larger proportion of variance than the latter. Moreover, they report that overall activation in lateral occipital cortex is best predicted by bias. 

Evaluation: This paper makes novel use of a new model that provides a good description of working memory performance in humans and monkeys. Determining the why and how of load-dependent changes in delay period activity is an important goal - dozens if not hundreds of studies have documented load-dependent changes in activation during WM task, yet a detailed explanation for these changes is lacking. I have two major concerns about this paper. The first is technical and focuses on whether it's fair for the authors to apply their DDM model to the current data set. The second is conceptual and focuses on what these data tell us about the structure of working memory or models thereof. I'm happy to clarify my comments and/or have a dialogue with the authors if it'd be helpful. 

Best wishes,

Edward Ester, Florida Atlantic University

Majors: 

1. Is it fair to apply the DDM to these data? As I understand it, the DDM used by the authors is constructed to explain bias and variance in subjects' responses under the assumption that to-be-reported items are uniformly sampled from a fixed parameter space - in this case, orientations on the interval [0,pi). Yet the orientations sampled here were preferentially drawn from subdistributions centered on 9 orientations (aside: was the jitter added to each orientation on each trial uniformly or normally distributed? I couldn't find this information in the manuscript). This creates two potential problems: the first and largest concern is that nonuniform sampling it could encourage bias in participants' orientation reports towards the nine "pedestal" orientations - e.g., participants are more likely to report a 23° stimulus as 20° because they've repeatedly seen exemplars drawn from a distribution centered on 20° - and this would result in biased parameter estimates. The second concern is that even if one assumes that participants' reports track the actual target values on each trial, it's not clear whether or how well the Panichello et al. model works in a nonuniform stimulus space. For example, does the model require uniform sampling? Does nonuniform sampling introduce additional bias or variance into parameter estimates? This absolutely needs to be worked out (and could presumably be done via simulation); otherwise it's very difficult to accurately interpret correlations between model parameters and BOLD data. 

2. Setting aside #1, I'm having a hard time understanding the implications of the study. The key points are that model estimates of variance correlate with load-dependent changes in the BOLD signal in frontoparietal cortex, and that model estimates of bias correlate with load-dependent changes in lateral occipital cortex. What are the implications of these findings? For example, some of the authors on this paper have previously advocated for a model where frontoparietal cortical areas coordinate memory storage in posterior sensory areas. On that account, I would've predicted frontoparietal areas to correlate more strongly with bias and visual areas to correlate with variance (e.g., frontoparietal areas implement bias to account for more variability in sensory areas during higher memory loads). As-is, I'm not sure what the current findings tell us about the architecture of working memory, other than that different brain areas correlate more strongly with different latent parameters. Some additional context and explanation is necessary. 

Minors: 

1. If space allows, it'd be helpful to include a brief introduction to the drift-diffusion model (DDM) in the main text and a mathematical derivation in the methods. I hadn't read the Panichello et al. paper prior to reviewing this study (though it's been on my "reading list" for a while), and the term "drift-diffusion model" has a storied history in the perceptual decision making literature (e.g., Ratcliff and others). Initially I thought the authors were going to be modeling forced-choice reaction time data and mapping those parameters on to BOLD activation, which led to some considerable head-scratching during my first read of the introduction and results. 

2. As Panichello et al. show, the DDM approach does a nice job of capturing biases in color report data. I'm less certain it generalizes to orientation, especially given that orientations weren't uniformly sampled in this experiment. For example, can the DDM capture well-known cardinal biases in delayed orientation reports (e.g., reported in multiple studies by Rademaker, Pratte, Landy, et al?). This strikes at the question of whether it's fair to model the current data with the DDM. 

Reviewer #2: Signed review by Ed Vogel

This article examines how a drift and diffusion model of within subject variation in working memory precision for orientation corresponds to delay period BOLD increases across cortex driven by item load (1 vs 3). The key finding is that while evidence for drift and diffusion were observed in frontal and parietal areas, diffusion explained most of the variance. By contrast, drift explained most of the variance in LO.

Overall, I think this is a very interesting and important study because it connects variation in WM behavioral precision to neural activity in a rigorous and model driven approach. The application of the DDM'd error performance to BOLD load activations is clever and novel. I found that the study narrative is clear and easy to follow and that the frontal and parietal results are very compelling. 

My chief concern is the interpretation of the LO results, most of which are "trending" in significance. I guess I'm not all that comfortable embracing the .08 as trending for drift while at the same time rejecting the .18 for diffusion. Of course, I think it is still important to report these results. However, I think some softening of the general conclusion that the results support a dichotomy is warranted. 

Perhaps related to the above point, I have some uncertainty about whether the smaller and shorted lived BOLD delay period signal in LO is impacting the fits to the behavioral models as compared to the larger longer lasting activation in PFC/PPC. More broadly, do increases in SNR (e.g., BOLD increase with load) equivalently impact the how the two models fare in predicting variance? 

Reviewer #3: This study examines behavioral and fMRI data from an experiment testing working memory for orientation. The behavioral responses are fit with a recently-proposed dynamical model of working memory based on drift-diffusion to stable attractors (Panichello et al., 2019) and correlations are identified between these parameters and BOLD actiivity in different brain regions.

This is a novel and (in principle) promising approach, but unfortunately the study has a major flaw: although the model is dynamical the data is not, i.e. experimental delay time is fixed and the BOLD correlation analysis is based on averaging over a fixed time window. Because the drift-diffusion model is fit to a single timepoint, the diffusion component presumably is simply measuring mean variability of recall errors and the drift component measuring the strength of stimulus-specific biases. Although only target-averaged response distributions are plotted (in Fig 1B) the pattern is consistent with a repulsive bias away from the cardinal orientations. This is a well-documented phenomenon for orientation judgments, not just in short-term memory (e.g. Taylor & Bays, J Neurosci, 2018) but also in immediate perceptual judgments (e.g. Tommasini et al, Vis Res, 2010) with no memory component and no delay interval - implying that temporal drift is unlikely to be responsible. It is also questionable whether difusion makes much of a contribution to error variability in this data, with some studies claiming the effects of delay are weak to non-existent (e.g. Shin, Zou & Ma, J Vis, 2017). Perhaps the authors could consider a complete reframing of the paper in terms of brain regions that correlate with (direct measures of) error variability vs biases.

---

## [Decision Letter · Decision Letter 2]

1 Jul 2020

Dear Dr Yu,

Thank you for submitting your revised Short Report entitled "Delay-period activity in frontal, parietal, and occipital cortex track different aspects of attractor dynamics in visual working memory" for publication in PLOS Biology. I have now obtained advice from the original reviewers and have discussed their comments with the Academic Editor. You will note that all reviewers --Edward Ester, Ed Vogel, and Paul Bays-- have signed their comments. 

Based on the reviews, we are positive about your manuscript. However, before committing to publication, we think you should modify the manuscript to address the remaining points raised by reviewer 3. In addition, the editors request changes to your title and abstract to make them more accessible to a general biology audience. See below for details. Please also make sure to address the data and other policy-related requests noted at the end of this email.

We expect to receive your revised manuscript within two weeks. 

***IMPORTANT INFORMATION FOR YOUR REVISION

Your revisions should address the specific points made by reviewer 3 by modifying your discussion.

In addition, we would like to you consider changing your title and modifying your abstract. I have written one suggestion here, but we'd be happy to discuss alternatives:

Title suggestion: "Imprecision in working memory can be explained by a combination of noise in frontoparietal cortex and stimulus-related biases in occipital cortex."

Regarding the Abstract, I generated this proposal. Please excuse any imprecision and feel free to improve it. The main idea is to make it more accesible by focusing on the high-level biological problem and then delving into the specifics.

Abstract suggestion: "Working memory is imprecise, and these imprecisions can be explained by diffusion from noise in the neural representation of memorized items and mitigated if memories are stored using a finite set of stable states known as discrete attractors. However, the neural activity that represents random diffusion and drift towards stable attractor states remains unknown. One important neural hallmark of working memory is persistent elevated delay-period activity in frontal and parietal cortex, and previous work has found that frontal and parietal delay-period activity correlates with the decline in behavioral memory precision observed with increasing memory load. Therefore, we set to investigate if frontoparietal and occipital delayed activity can explain different aspects of error sources in working memory. We analyzed data from an existing experiment in which subjects performed delayed recall for line orientation, at different loads, during fMRI scanning. We modeled subjects’ behavior using a discrete attractor model and calculated within-subject correlation between frontal and parietal delay-period activity and the drift and diffusion sources of memory error. We found that although increases in frontal and parietal activity were associated with increases in both diffusion and drift, diffusion explained the most variance in frontal and parietal delay-period activity. In comparison, a subsequent whole-brain regression analysis showed that drift rather than diffusion explained the most variance in delay-period activity in lateral occipital cortex. These results thus show that imprecision in working memory can be caused by a combination of effects of noise in frontoparietal cortex and of stimulus-related biases in occipital cortex."

Please submit the following files along with your revised manuscript:

In addition to the remaining revisions and before we will be able to formally accept your manuscript and consider it "in press", we also need to ensure that your article conforms to our guidelines. A member of our team will be in touch shortly with a set of requests. As we can't proceed until these requirements are met, your swift response will help prevent delays to publication.

*Copyediting*

*Published Peer Review History*

*Early Version*

*Submitting Your Revision*

Sincerely,

Gabriel Gasque, Ph.D., 

Senior Editor

PLOS Biology

ETHICS STATEMENT:

-- Please indicate within your manuscript the ID number of the protocols approved by the University of Wisconsin–Madison Health Sciences Institutional Review Board. 

-- Please indicate within your manuscript if your protocols approved by the University of Wisconsin–Madison Health Sciences Institutional Review Board adhered to the Declaration of Helsinki or any other national or international ethical guidelines. 

DATA POLICY:

Note that we do not require all raw data. Rather, we ask for all individual quantitative observations that underlie the data summarized in the figures and results of your paper. For an example see here: http://www.plosbiology.org/article/info%3Adoi%2F10.1371%2Fjournal.pbio.1001908#s5

These data can be made available in one of the following forms:

Regardless of the method selected, please ensure that you provide the individual numerical values that underlie the summary data displayed in the following figure panels: Figures 1B-C, 2A-D, 3A-E, S1, S2, S3, S4, S5, S6A-D, S7A-D, and S8A-D.

Please also ensure that each figure legend in your manuscript include information on where the underlying data can be found and ensure your supplemental data file/s has a legend.

Reviewer remarks:

Reviewer #1, Edward Ester: The authors have addressed my earlier comments re: the modeling approach to my satisfaction, and have done an admirable job responding to other reviewers' comments (including several new analyses and an analysis of third-party data to support their arguments). I have no further concerns and congratulate the authors on a nice study!

Reviewer #2: I am happy with this version of the manuscript. The authors did a nice job of responding to the prior round of reviews. I have no remaining concerns. I feel that this work will have a substantial impact on the field.

Ed Vogel

Reviewer #3, Paul Bays: See attachment.

---

## [Editor Report · Decision Letter 3]

10 Aug 2020

Dear Dr Yu,

On behalf of my colleagues and the Academic Editor, Frank Tong, I am pleased to inform you that we will be delighted to publish your Short Reports in PLOS Biology. 

Early Version

PRESS 

Kind regards,

Alice Musson

Publishing Editor, 

PLOS Biology

on behalf of

Gabriel Gasque,

Senior Editor

PLOS Biology